# Polymer-acid-metal quasi-ohmic contact for stable perovskite solar cells beyond a 20,000-hour extrapolated lifetime

Junsheng Luo[1,2,3,7], Bowen Liu[2,7], Haomiao Yin[1], Xin Zhou[4], Mingjian Wu [4], Hongyang Shi[2], Jiyun Zhang [2,5], Jack Elia [2], Kaicheng Zhang[2], Jianchang Wu[2,5], Zhiqiang Xie[2], Chao Liu[2,5], Junyu Yuan[3], Zhongquan Wan [1,3] ✉, Thomas Heumueller [2,5], Larry Lüer [2,5], Erdmann Spiecker [4], Ning Li [2,5,6], Chunyang Jia[1,3] ✉, Christoph J. Brabec [2,5] ✉ & Yicheng Zhao [1,5] ✉

The development of a robust quasi-ohmic contact with minimal resistance, good stability and cost-effectiveness is crucial for perovskite solar cells. We introduce a generic approach featuring a Lewis-acid layer sandwiched between dopant-free semicrystalline polymer and metal electrode in perovskite solar cells, resulting in an ideal quasi-ohmic contact even at elevated temperature up to 85 °C. The solubility of Lewis acid in alcohol facilitates nondestructive solution processing on top of polymer, which boosts hole injection from polymer into metal by two orders of magnitude. By integrating the polymer-acid-metal structure into solar cells, devices exhibit remarkable resilience, retaining 96% ± 3%, 96% ± 2% and 75% ± 7% of their initial efficiencies after continuous operation in nitrogen at 35 °C for 2212 h, 55 °C for 1650 h and 85 °C for 937 h, respectively. Leveraging the Arrhenius relation, we project an impressive $T_{80}$ lifetime of 26,126 h at 30 °C.

Perovskite solar cells (PSCs) have exhibited remarkable power conversion efficiencies (PCEs) surpassing 26%[1]. However, today's potential of PSCs for practical applications is limited by their longevity issues[2]. Significant research efforts have been directed towards stabilizing the light-absorbing materials as well as their interfacial layers. These efforts encompassed various aspects, such as the mixed-cation mixed-halide perovskite[3–11], the electron-transporting layer (ETL)[12–15], the hole-transporting layer (HTL)[16–22], the perovskite/ETL interface[23–25] and the perovskite/HTL interface[26–29], all aimed at enhancing the durability of PSCs. For example, Yang et al. used theophylline to passivate the

surface defects of perovskite crystals in PSCs, and the resulting device exhibits 10% degradation of device performance under continuous illumination at 40 °C for 500 h[29]. Green et al. have incorporated 1-dodecanethiol into Spiro-OMeTAD-based HTL and produced PSCs that maintained 90% of peak efficiency under continuous illumination for 1000 h at 30–35 °C[18]. Nevertheless, despite multiple gradual achievements, these accomplishments still fall short to meet the longevity standards necessary for outdoor commercialization.

Recently, it has become increasingly clear that the stability of quasi-ohmic contacts plays a significant role in determining the overall

[1]National Key Laboratory of Electronic Films and Integrated Devices, School of Integrated Circuit Science and Engineering, University of Electronic Science and Technology of China, 611731 Chengdu, PR China. [2]Institute of Materials for Electronics and Energy Technology (i-MEET), Department of Materials Science, Friedrich-Alexander University Erlangen-Nürnberg, Martensstr. 7, 91058 Erlangen, Germany. [3]Shenzhen Institute for Advanced Study, University of Electronic Science and Technology of China, 518110 Shenzhen, PR China. [4]Institute of Micro- and Nanostructure Research & Center for Nanoanalysis and Electron Microscopy (CENEM), Department of Materials Science, FriedrichAlexander-Universität Erlangen-Nürnberg, Cauerstr. 3, D-91058 Erlangen, Germany. [5]Helmholtz-Institute Erlangen-Nürnberg (HI-ERN), Immerwahrstr. 2, 91058 Erlangen, Germany. [6]Institute of Polymer Optoelectronic Materials and Devices, State Key Laboratory of Luminescent Materials and Devices, South China University of Technology, 510640 Guangzhou, PR China. [7]These authors contributed equally: Junsheng Luo, Bowen Liu. ✉e-mail: zqwan@uestc.edu.cn; cyjia@uestc.edu.cn; christoph.brabec@fau.de; zhaoyicheng@uestc.edu.cn

stability of PSCs, especially under elevated temperatures[30–34]. Luther et al. found that high-vacuum-deposited $MoO_x$/metal quasi-ohmic contact is unstable, as illumination at 70 °C induced morphological changes in the $MoO_x$ layer, resulting in rapid efficiency losses within initial 100 h[30]. Similarly, we revealed the instability of Ta-$WO_x$/metal contact in PSCs under illumination and heat (60 °C), resulting in rapid fill factors (FF) losses of PSCs[34]. More recently, it has been discovered that ionic-salt-doped HTL forms robust quasi-ohmic contact, demonstrating high photo-thermal-operational stability in these devices[35,36]. Subsequently, we also developed a bilayer-conducting-polymer architecture comprising a doped $p$-type poly[bis(4-phenyl)(2,4,6-tri-Methylphenyl)amine] (PTAA) on top of an otherwise undoped polymer poly[5,5′-bis(2-butyloctyl)-(2,2′-bithiophene)−4,4′-dicarboxylate-alt-5,5′−2,2′-bithiophene] (PDCBT) to form a quasi-ohmic contact with a metal electrode (i.e., Au)[34]. This device design sustains a consistent power output over 1450 h of continuous operation at 60 °C. Despite the remarkable stability, that bilayer architecture still has two limitations. To begin with, the significant similarity in solubility of the two polymers gives rise to a processing challenge in which the upper polymer tends to dissolve the underlying polymer. The problem of dissolution issue can affect the film crystallinity and contact quality, imparting a sensitivity to processing conditions and the surrounding atmosphere. This, in turn, can have a direct impact on the consistency and reliability of efficiency and stability. Second, the excess amount of a second conducting polymer layer raises the overall cost of the devices—a aspect that requires careful consideration within the framework of techno-economical analysis or life cycle assessment (LCA) or life cycle cost analysis (LCCA).

Here, we introduce an inherently robust quasi-ohmic contact design, which is based on a polymer-acid molecule-metal bilayer structure forming a generic heterojunction contact to the perovskite layer. The $p$-type, dopant-free semicrystalline polymer and the acid molecule are dissolved in orthogonal solvents, ensuring consistent reproducibility a negligible batch-to-batch variability. The Lewis acids are significantly cheaper than alternative conducting polymers used for bilayer architectures. More importantly, a thin layer of Lewis-acid molecules creates an ideal quasi-ohmic contact between the semicrystalline polymer and the metal, resulting in high FF for PSCs. Accordingly, the champion PSC exhibits excellent long-term photo-thermal-operational stability under continuous illumination at temperatures of 35 °C ($T_{99} = 2212$ h), 55 °C ($T_{98} = 1650$ h) and 85 °C ($T_{82} = 937$ h) in $N_2$ atmosphere.

## Results and discussion
### Evolution of device architecture
The research progress and evolution of dopant-free semicrystalline polymer-based PSCs architecture regarding quasi-ohmic contacts by our group are shown in Fig. 1a. The conventional dopant-free HTL architecture of PSCs consists of transition metal oxide (e.g., $MoO_x$, Ta-$WO_x$) sandwiched between a semicrystalline polymer and a metal electrode, as depicted in Fig. 1a[33]. However, having identified the thermal instabilities of metal oxides, we adopted a doped conducting polymer to establish a more robust quasi-ohmic contact, as illustrated in a schematic band diagram in Fig. 1a, b[34]. We further argue that a thin layer of a Lewis-acid molecule at the surface does enable the formation of an efficient quasi-ohmic contact, assuming that the surface doping can dominate the carrier density in such thin polymer layers. Such $p$-type doping causes a downshift of the work function (WF) of the underlying semicrystalline polymer, and, finds an equilibrium by transferring electrons or holes (Fig. 1b & Supplementary Fig. 1). To verify the interface-relevant processes, we selected a series of $p$-type semicrystalline polymers (Fig. 1c) that are dissolved in non-polar solvent such as orthodichlorobenzene (ODCB), as well as Lewis-acid molecule tris(pentafluorophenyl)borane (BCF) and its derivatives (Fig. 1d), i.e., lithium tetrakis(pentafluorophenyl)borate-ethyl ether

complex (Li-BCF), triphenylmethylium tetrakis(pentafluorophenyl) borate (C-BCF), N,N-dimethylanilinium tetrakis(pentafluorophenyl) borate (N-BCF) and 4-isopropyl-4′-methyldiphenyliodonium tetrakis(pentafluorophenyl)borate (I-BCF) that are dissolved in polar solvent such as isopropanol (IPA). Different bilayer combinations of these semicrystalline polymers with alcohol-soluble Lewis acid BCF and its derivatives were investigated to explore contact formation for the polymer-acid-metal structure in PSCs.

### Quasi-ohmic-contact property of the polymer-acid-metal structure
Hole-only devices with the architecture ITO/PEDOT:PSS/PDCBT/interlayer/Au were fabricated to investigate the hole injection capability from the metal into the semicrystalline polymer. As shown in Fig. 2a, the current injected from the Au electrode into the PDCBT layer under forward bias is lower than that injected from PEDOT:PSS under reverse bias. That evidences a hole injection barrier at the PDCBT/Au interface. Hole injection into PDCBT is improved when a transition metal oxide (i.e., Ta-$WO_x$) is inserted between PDCBT and Au, though a residual barrier still persists. In contrast, the hole injection current is considerably improved by two orders of magnitude when processing the alcohol-soluble Lewis acid (e.g., BCF) as interlayer. We extended this approach to various combinations of $p$-type semicrystalline polymers with BCF. A significant improvement of current injection from the Au electrode is observed for nearly all combinations, demonstrating a generic approach to forming a quasi-ohmic contact (Supplementary Fig. 2).

Interface contact was further studied based on the vertical architecture of ITO/PDCBT/interlayer/Au, which shows a stark contrast between metal oxide and Lewis acid interface modifiers. Under negative bias, the hole injection current is inhibited by an injection barrier from ITO into PDCBT (Fig. 2b). Spin coating various BCF derivatives effectively dopes the bulk of semicrystalline polymer and thus mitigates both injection barriers, the larger one from ITO to polymer and the smaller one from Au to polymer (Supplementary Fig. 3). The noticeable bulk doping results in enhanced device reproducibility (Supplementary Fig. 4). This observation suggests that Lewis acid and metal oxide doping employ distinct mechanisms in establishing quasi-ohmic contacts. The doping effect induced by Lewis acids appear to lead to a bulk doping effect that lowers the $p$-type Fermi level closer to the valence band. That reduces the WF difference to both metal electrodes (ITO, Au). As the work function of ITO is larger than the one of Au, the effect is stronger expressed for the injection from the ITO side. We further demonstrate that BCF can universally reduce the energy barrier and improve the injection current between ITO or Au and various $p$-type semicrystalline polymers (Supplementary Fig. 5).

To evidence the doping effect, we further measure lateral conductances of various polymers with and without Lewis-acid molecules. Two-terminal current-voltage characteristics show linear response for the pristine PDCBT-based devices (Fig. 2c). After spin-coating Ta-$WO_x$ or BCF from orthogonal IPA solvent, the current is improved by one and two orders of magnitude, respectively, and shows good reproducibility (Supplementary Fig. 6). Similar results are observed in other $p$-type polymers (Supplementary Fig. 7), confirming the strong doping effect by the Lewis acids. We also investigate the effect of BCF derivatives on the lateral conductance of PDCBT, and found effective doping and overall improved lateral conductance (Supplementary Fig. 8).

The distribution of the Lewis-acid molecules across the bulk is investigated by time-of-flight secondary ion mass spectrometry (TOF-SIMS), as shown in Fig. 2d and Supplementary Fig. 9. The $F^-$ and $CF^-$ signals, representing BCF, showcase a higher concentration enrichment at the surface of the PDCBT layer, followed by a gradual decrease across the polymer bulk. This distribution profile indicates a gradient doping within PDCBT featuring surface enrichment and subsequent bulk diffusion (Fig. 1b). The ensuing gradient doping explains the

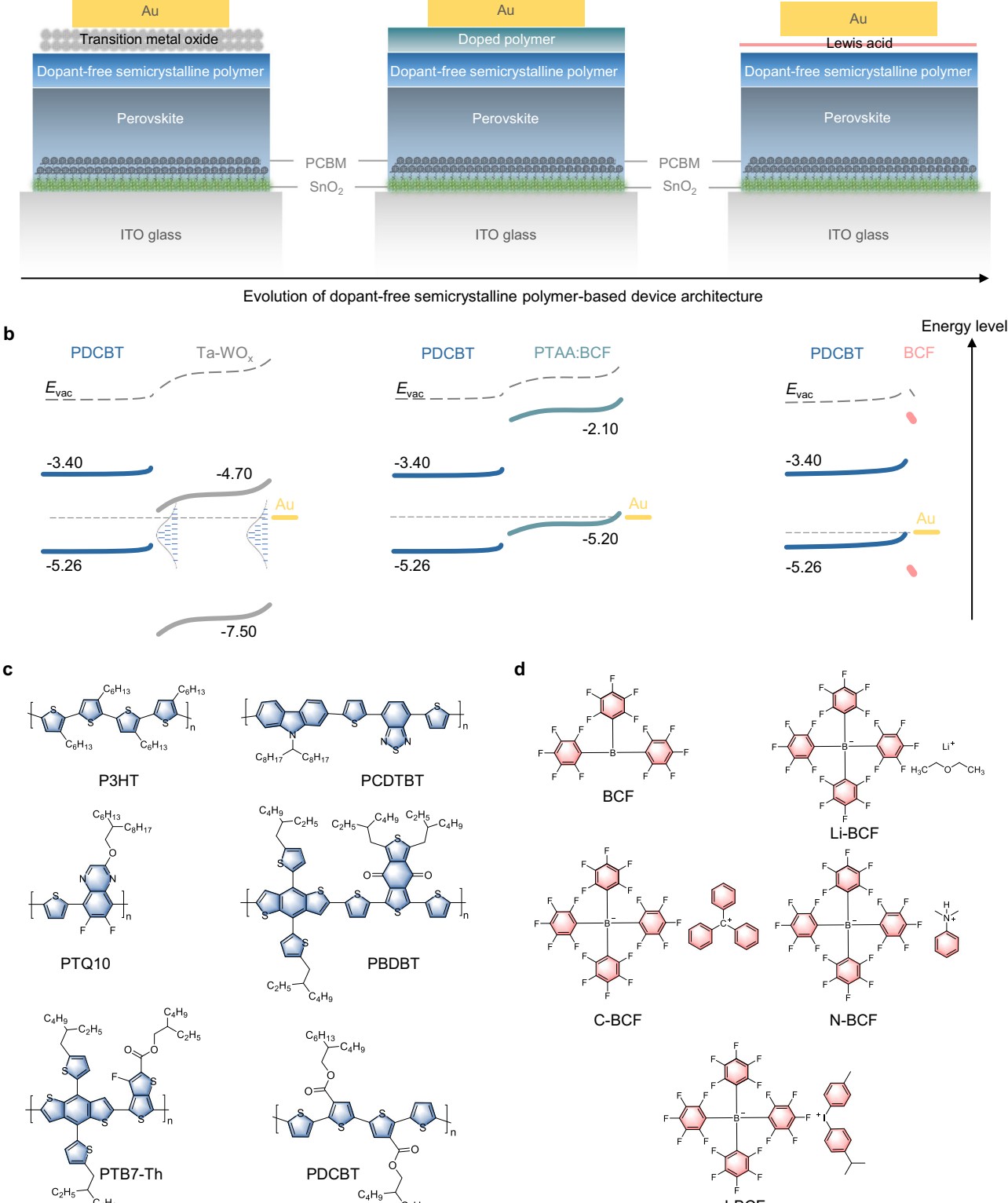

**Fig. 1 | Device architecture of PSCs, energy diagram of quasi-ohmic contacts, and molecular structure of *p*-type semicrystalline polymers and alcohol-soluble Lewis acid BCF and its derivatives. a** Device architecture with *p*-type semicrystalline polymer/transition metal oxide/Au, *p*-type semicrystalline polymer/doped polymer/Au, and *p*-type semicrystalline polymer/alcohol-soluble Lewis acid/ Au as quasi-ohmic contacts. **b** Energy diagram of PDCBT/Ta-WO$_x$/Au, PDCBT/PTAA:BCF/Au, and PDCBT/BCF/Au quasi-ohmic contacts. **c**, **d** Chemical structure of (**c**) *p*-type semicrystalline polymers and (**d**) alcohol-soluble Lewis acid BCF and its derivatives.

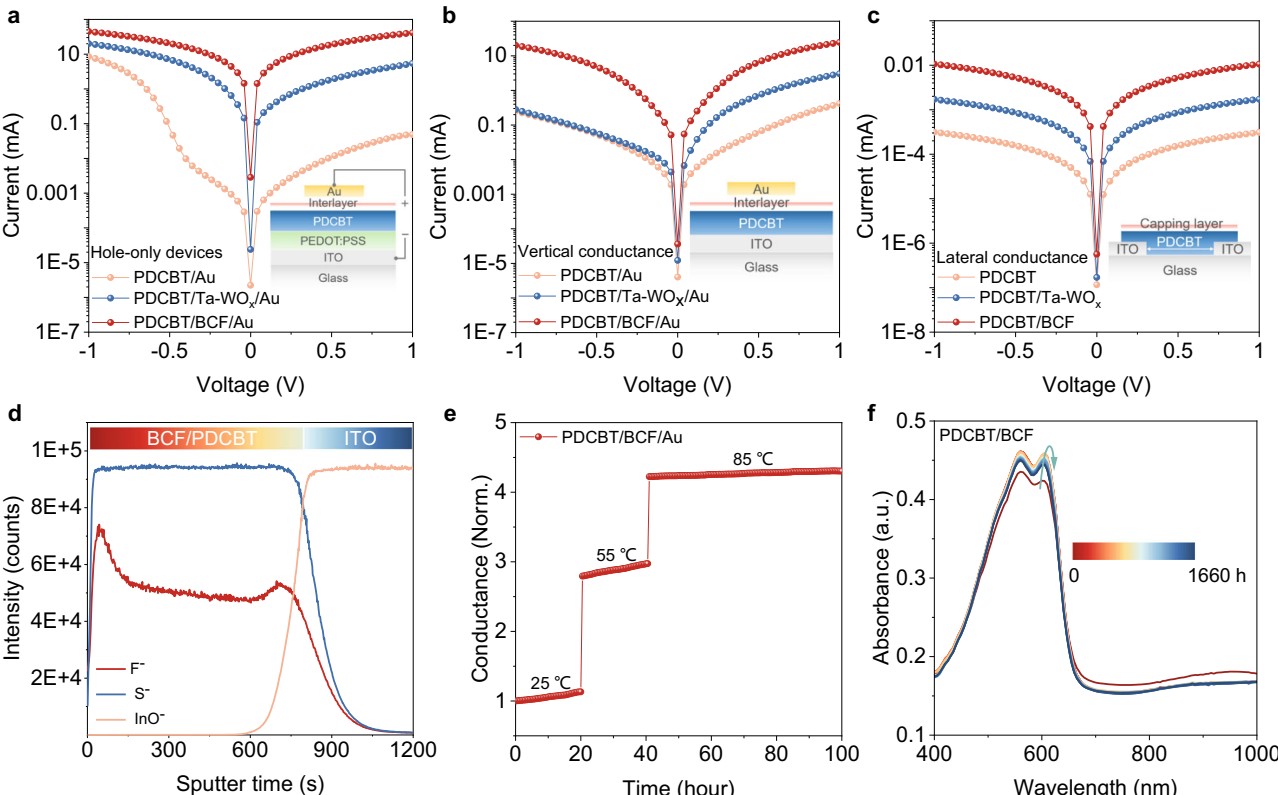

**Fig. 2 | Quasi-ohmic contacts engineering. a** Current-voltage characteristics of hole-only device ITO/PEDOT:PSS/PDCBT/interlayer/Au (inset). **b** The vertical conductance measurements in an architecture consisting of glass/ITO/PDCBT/interlayer/Au (inset). **c** The lateral conductance measurements in a structure consisting of interdigitated ITO/functional layer/interdigitated ITO (inset). **d** TOF-SIMS depth profile of ITO/PDCBT/BCF. **e** The vertical conductance stability for the glass/ITO/PDCBT/BCF/Au at elevated temperature in a $N_2$-filled chamber. **f** In-situ UV-vis absorption property of the encapsulated glass/PDCBT/BCF during the aging process at 55 °C for 1660 h. Source data are provided as a Source Data file.

enhanced hole injection from Au into PDCBT as well as ITO into PDCBT. The doping effect is also consistent with the downshift of *WF* detected by Kelvin probe (Supplementary Fig. 1), though this method is more surface-sensitive than TOF-SIMS. The described scenario is applicable universally for other BCF derivatives as well (Supplementary Fig. 1).

Next, we perform the thermal stability of these quasi-ohmic contacts (Fig. 2e). In contrast to the conductance instability with the PDCBT/Ta-WO$_x$/Au structure, PDCBT/BCF/Au shows ultra-stable conductance property between 25 and 85 °C (Supplementary Fig. 10). We track the evolution of in-situ UV-vis absorption spectra of the PDCBT/BCF during aging at 55 °C for 1660 h, as shown in Fig. 2f. The absorption intensity of PDCBT/BCF is increased at the beginning of aging, which is attributed to minor morphological changes of the BCF redistribution near the surface of PDCBT[37]. A similar absorption enhancement was observed in other all-organic bulk-doped polymers. However, it was not observed for pristine PDCBT, PDCBT/Ta-WO$_x$, and PDCBT/MoO$_x$ bilayers (Supplementary Fig. 11). The subsequent slow decrease in absorption intensity indicates packing disorder of polymer chain, this phenomenon also observed in the pristine PDCBT layer. Although the detailed mechanism is still a puzzle to us, the slow decrease did not affect the vertical conductance stability in the devices. These results indicate that polymer-acid-metal quasi-ohmic contact has the potential to produce efficient and stable PSCs.

## Device performance of PSCs with polymer-acid-metal contacts

We integrate the polymer-acid-metal quasi-ohmic contact into n-i-p devices with an architecture of glass/ITO/SnO$_2$/PCBM/perovskite/polymer-acid-metal. The contact involves various combinations of dopant-free semicrystalline polymers as HTLs, including P3HT/PCDTBT/PTQ10/PBDBT/PTB7-Th/PDCBT, coupled with alcohol

processable Lewis acid BCF and its derivatives (Li-BCF/C-BCF/N-BCF/I-BCF). The device architecture, illustrated in Fig. 3a, is corroborated through cross-sectional scanning transmission electron microscopy (STEM) imaging. The elemental mapping, complemented by energy-dispersive X-ray spectroscopy (EDS), elucidates the distinct layers comprising the device architecture. The stoichiometry of the perovskite used in this study is $Cs_{0.05}FA_{0.85}MA_{0.1}Pb(I_{0.95}Br_{0.05})_3$, which was chosen among 160 compositions using a high-throughput robotic platform according to stability criteria[34]. MgF$_2$ is deposited on top of Au electrode to protect the device.

Upon direct contact with the metal electrode, PSCs based on the undoped polymers exhibit a characteristic *S*-shaped current density-voltage (*J-V*) curves, leading to low PCEs under AM 1.5 G simulated illumination (Fig. 3b & Supplementary Fig. 12). The *S*-shape characteristic is a signature of an energy barrier at the interface or critically low conductivity of the transport layer. With an interlayer of solution-processed BCF layer between the metal and the pristine polymer transport layer, the *S*-shape in the *J-V* curves disappear for all the polymer-based PSCs, indicating the transition from a Schottky to quasi-ohmic contact. This transition is observed for various BCF derivatives with PDCBT (Fig. 3c & Supplementary Fig. 13). Among these combinations, device employing PDCBT with BCF delivers the best performance, yielding a PCE of around 21% with negligible hysteresis (Fig. 3d). The steady-state PCE of 21.31% is close to 21.33% in reverse direction (Fig. 3e). The difference between the integrated short-circuit current density ($J_{SC}$) value from the external quantum efficiency (EQE) spectrum and the $J_{SC}$ value (23.07 mA cm$^{-2}$) obtained from *J-V* measurement is about 3%, which is negligible (Fig. 3f). Importantly, the implementation of the polymer-acid-metal architecture (PDCBT/BCF/Au) significantly mitigates batch-to-batch and substrate-to-substrate

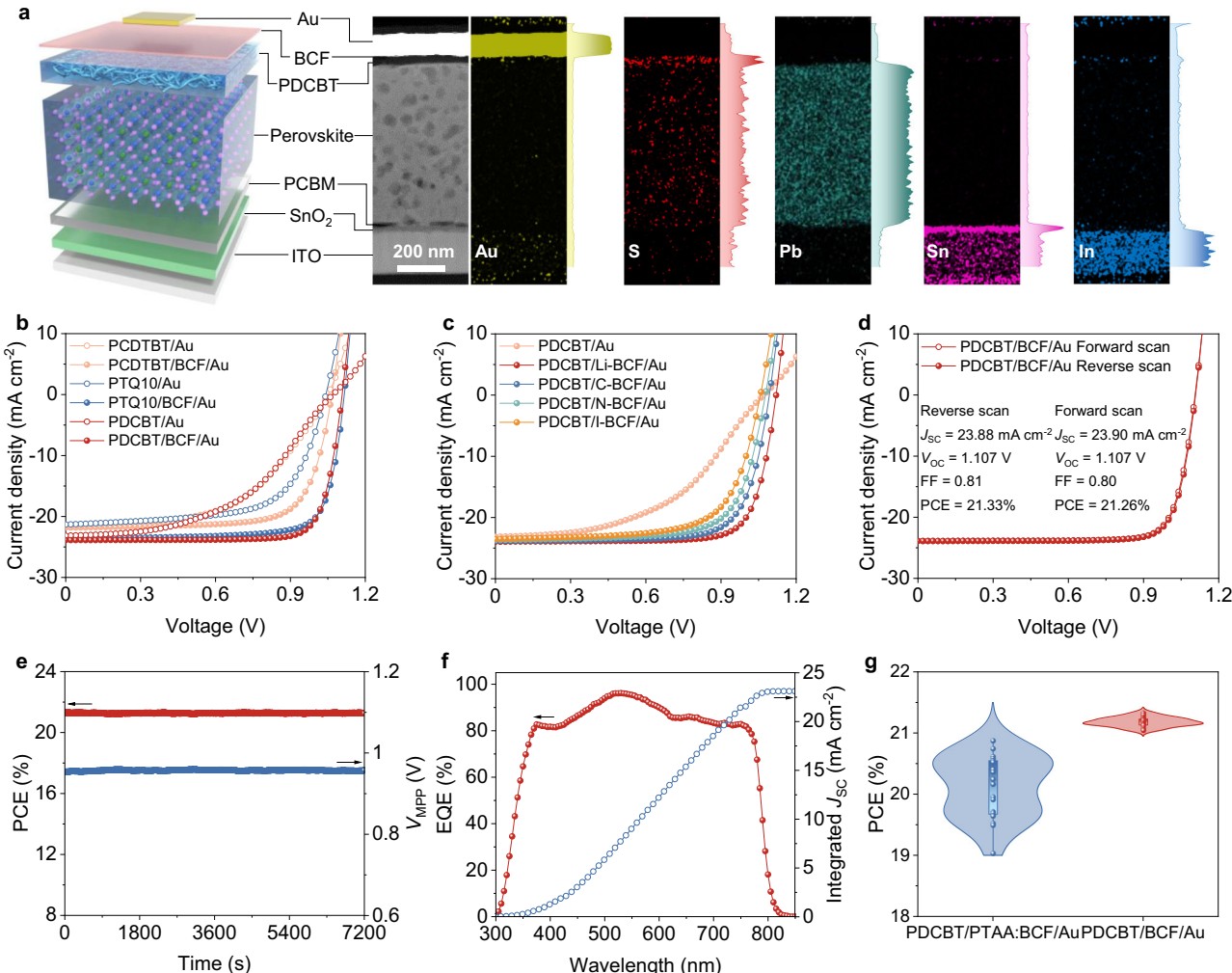

**Fig. 3 | Device performances. a** Schematic illustration of PSC architecture, cross-sectional STEM z-contrast image, and the corresponding EDS mapping of device. **b** *J*-*V* curves with reverse scan of the champion PSCs based on the polymer-acid-metal structure using various *p*-type semicrystalline polymers as HTLs and with or without BCF as quasi-ohmic contact layer. **c** *J*-*V* curves with reverse scan of the champion PSCs with polymer-acid-metal structure using various alcohol-soluble BCF derivatives as quasi-ohmic contact layer. **d** *J*-*V* curves of the champion cell based on the structure of PDCBT/BCF/Au with forward and reverse scan. **e** Steady-state output of the champion device (PDCBT/BCF/Au) with maximum power point (MPP) tracking under simulated AM1.5 G illumination. **f** EQE spectrum and integrated $J_{SC}$ of the champion PSC with the structure of PDCBT/BCF/Au. **g** The statistics of PCE values obtained from *J*-*V* characteristics for devices based on PDCBT/PTAA:BCF/Au and PDCBT/BCF/Au structures. Source data are provided as a Source Data file.

variations (Fig. 3g), as evidenced through a massive reduction in performance variation as compared to the bilayer-conducting polymer structure (PDCBT/PTAA:BCF/Au). The improvement in reproducibility is attributed to the utilization of orthogonal solvents for the polymer-acid-metal structure. The statistical distribution of other photovoltaic parameters is summarized in Supplementary Fig. 14. Thus, we successfully demonstrate an ideal quasi-ohmic contact with low cost (Supplementary Table 1) and high reproducibility by using a polymer-acid-metal architecture, providing an attractive alternative to commonly used metal oxides and doped conducting polymers. Meanwhile, the performance of device with an architecture of ITO/SnO₂/PCBM/ perovskite/PTAA/BCF/Au was also studied. In stark contrast to semicrystalline PDCBT film with efficient hole transporting, the amorphous PTAA film requires bulk doping to realize hole transporting. As a result, the device based on PTAA/BCF/Au architecture shows a low FF, as shown in Supplementary Fig. 15.

## Photo-thermal-operational stability of PSCs

To rigorously assess the operational stability of PSCs featuring the polymer-acid-metal structure (PDCBT/BCF/Au), we adopt the International Summit on Organic Photovoltaic Stability (ISOS) protocol ISOS-L-2, which entails combined light and heat stress[38]. Following ISOS-L-2 protocols, the evolution of device degradation is aged at the MPP with different temperatures (35, 55, and 85 °C) under continuous light illumination in homemade chambers with continuous flowing N₂. A bunch of *J*-*V* curves were acquired through both reverse and forward scan directions (Fig. 4), along with the corresponding evolution of PCEs tracked during the aging period. There is negligible hysteresis in the *J*-*V* curves over the entire aging period (Fig. 4a–f). The trend of normalized PCEs over time are shown in Fig. 4g, h after calibrating for light intensity perturbations using a standard silicon photodetector as reference. Five individual devices were measured under each condition. The evolution of other photovoltaic parameters $J_{SC}$, $V_{OC}$, and FF as a function of aging time are summarized in the Supplementary Figs. 16–27. PDCBT/Ta-WOₓ/Au reference architecture degraded under 55 °C and 85 °C are presented in Supplementary Figs. 28 and 29. Notably, the PSCs featuring the polymer-acid-metal PDCBT/BCF/Au structure demonstrate good long-term stability. The devices retain 96% ± 3%, 96% ± 2%, and 75% ± 7% of their initial PCEs after continuous illumination at 35 °C for 2212 h, 55 °C for 1650 h and 85 °C for 937 h

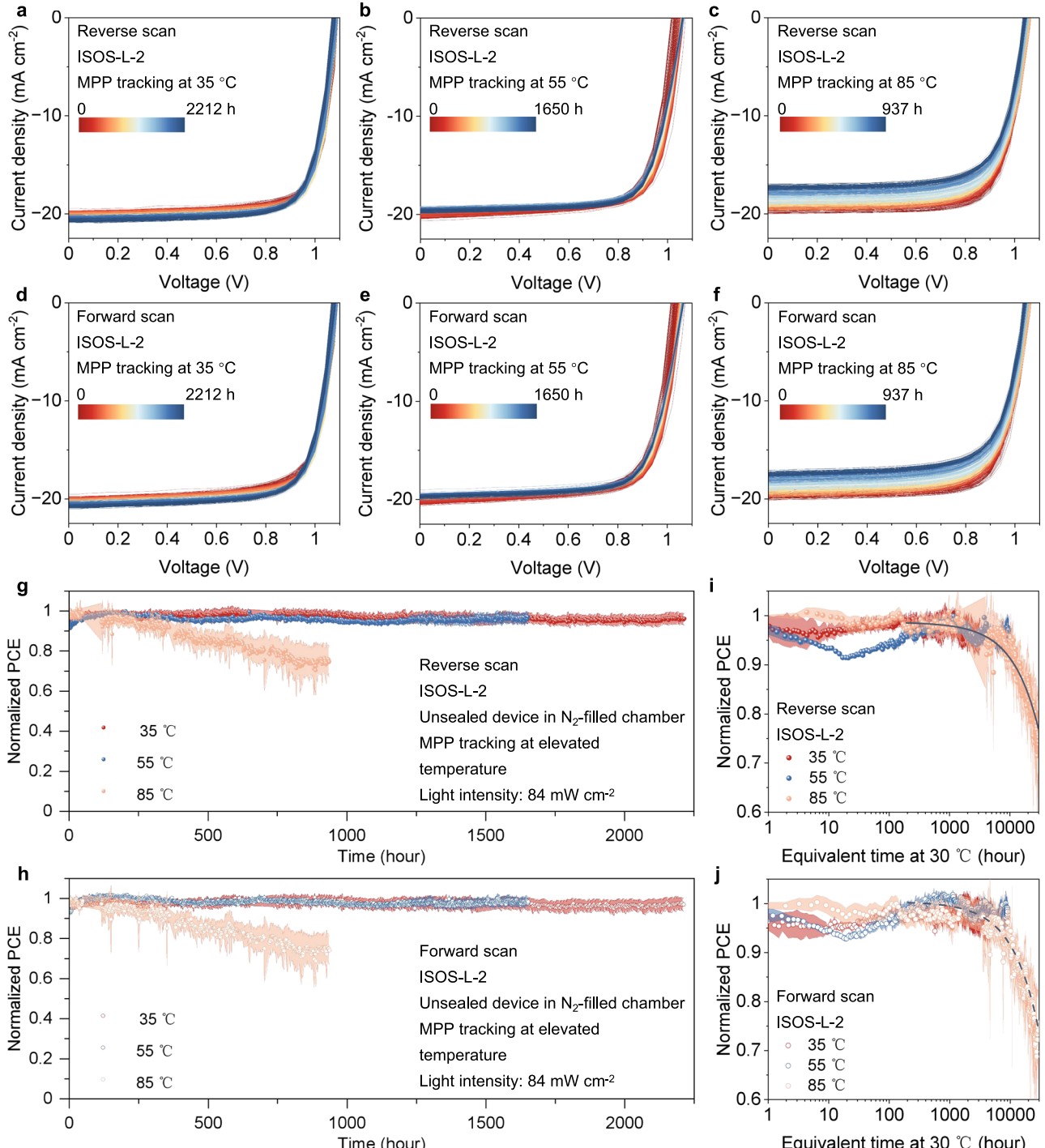

**Fig. 4 | Operational stability of PSCs. a–f** *J-V* curves for the representative devices based on PDCBT/BCF/Au structure aged at **a**, **d** 35 °C for 2212 h, **b**, **e** 55 °C for 1650 h and **c**, **f** 85 °C for 937 h in N₂-filled chambers under continuous light illumination (84 mW cm⁻²) without encapsulation in reverse and forward direction, respectively. **g**, **h** Long-term stability of PSCs based on PDCBT/BCF/Au structure, data points originated from **g** reverse *J-V* scan and **h** forward *J-V* scan, respectively.

**i**, **j** Normalized PCE originated from **i** reverse *J-V* scan and **j** forward *J-V* scan of PSCs with PDCBT/BCF/Au structure plotted against the equivalent aging time at 30 °C by leveraging the Arrhenius relation. All of the error bars in Fig. 4g, h, i, and j represent the standard deviations from 5 individual devices for each temperature. Source data are provided as a Source Data file.

with negligible *J-V* hysteresis. We notice that the degradation of PCE at 85 °C is mainly caused by the decline in $J_{SC}$ and FF (Supplementary Figs. 24–27), which may be attributed to an instability of the Au electrode (Supplementary Fig. 30).

The degradation rate *k*, defined in the formula of PCE(*t*) = *kt* + *Const.*, is intrinsically associated with the aging temperature of the devices. An Arrhenius model is used to estimate the activation energy

associated with degradation by utilizing the degradation rate *k*, as shown in Eq. (1), where $E_a$ is the activation energy of degradation, *A* is constant and $k_B$ is Boltzmann's constant[39–43].

$$k(T) = A\exp\left(\frac{-E_a}{k_B T}\right) \qquad (1)$$

According to Eq. (1), the activation energy is equivalent to the slope:

$$E_a = -\frac{\partial \ln(k(T))}{\partial\left(\frac{1}{k_B T}\right)} \tag{2}$$

Supplementary Fig. 31 shows the temperature-dependent degradation rate as a function of 1/T, the fitted activation energy is $0.61 \pm 0.11$ eV and $0.59 \pm 0.10$ eV for the data extracted from reverse and forward J-V scan, respectively. The activation energy for our n-i-p structure is close to that reported in p-i-n structure with a similar perovskite composition, implying that perovskite decomposition determines the temperature-dependent degradation rate in devices[44]. A pivotal attribute commonly used in longevity models is the acceleration factor (AF), quantifying the ratio of the accelerated degradation rate at elevated temperature ($k_{acc}$) to its counterpart at a standard operating condition ($k_{ref}$). This relationship is expressed in Eq. 3[43]:

$$AF = \frac{k_{acc}}{k_{ref}} = \exp\left(\frac{E_a}{k_B}\left[\frac{1}{T_{ref}} - \frac{1}{T_{acc}}\right]\right) \tag{3}$$

where $T_{ref}$ and $T_{acc}$ are the operating temperatures during aging at standard and accelerated operating conditions, respectively.

We calculate the AF for distinct elevated temperatures, specifically 35 °C, 55 °C and 85 °C, as illustrated in Supplementary Fig. 32. Utilizing the AF values at elevated temperatures, we proceed to calculate the equivalent operational lifetime under standard condition ($T_{ref}$ = 30 °C) by multiplying aging time with the AF. The time required to decay 20% of the initial efficiency is defined as $T_{80}$, which serves as a figure of merit for cell stability. By fitting the degradation data, we estimate the $T_{80}$ lifetime up to 26,126 and 22,795 h at 30 °C for PSCs based on polymer-acid-metal structure (PDCBT/BCF/Au) in reverse and forward J-V scan, respectively (Fig. 4i, j). This longevity holds the potential to satisfy the market demand for highly stable PSCs[45]. To make comparisons with the latest available results, we tabulate the long-term stability data of PSCs from the literatures (Supplementary Table 2), with specific the aging condition, the cell structure, and the efficiency loss. The stability of our optimized devices is superior to previous reports, under the combined stress of light and heat, especially for the n-i-p architecture. We highlight the critical importance of achieving a robust quasi-ohmic contact between the HTL and the Au electrode. Replacing the interfacial Ta-WO$_x$ with the acidic molecule BCF increased the $T_{80}$ lifetime at 85 °C from few hours to about 1000 h.

In summary, we report n-i-p PSCs with good stability under light and elevated temperature through the development of a robust quasi-ohmic contact between the p-type transport layer and the metal electrode. By combining a p-type semicrystalline polymer with alcohol-soluble Lewis acid to engineer all-organic doped HTLs, we achieved PSCs with elevated efficiency, cost-effectiveness, reliable reproducibility, and enhanced operational stability. Notably, PSCs featuring a polymer-acid-metal architecture with PDCBT/BCF/Au exhibit a good degradation resistance at enhanced operational temperatures, retaining 96% ± 3%, 96% ± 2%, and 75% ± 7% of their initial efficiencies during exposure to temperatures of 35 °C for 2212 h, 55 °C for 1650 h and 85 °C for 937 h, respectively. Our innovative approach also provides a generic solution for the formation of quasi-ohmic contact for other advanced solution-processed optoelectronic devices, such as perovskite or organic light-emitting diodes, field-effect transistors or photodetectors. Our findings make a significant contribution by providing a key method for the formation of temperature-stable low-loss quasi-ohmic contacts between p-type polymers and metal electrodes, thus enhancing the performance and reliability of various optoelectronic technologies.

## Methods

### Materials and solution preparation

SnO$_2$ colloid precursor was purchased from Alfa Aesar. PCBM was purchased from Nano-C. PbI$_2$, PbBr$_2$, BCF, and BCF derivatives were purchased from TCI. FAI and MABr were ordered from Greatcell Solar. CsBr was purchased from Xi'an P-OLED. PDCBT was purchased from 1-Materials. Ta-WO$_x$ colloidal solution was purchased from Avantama Ltd (product number: 5030). PEDOT:PSS (VPAI 4083) was purchased from Heraeus Clevios. IPA, N,N-dimethylformamide (DMF), dimethyl sulfoxide (DMSO), chlorobenzene (CB), and ODCB were purchased from Sigma-Aldrich. The SnO$_2$ solution was prepared by diluting 15 wt% SnO$_2$ aqueous solutions (500 μL) with 2.5 mL of IPA/H$_2$O (1:1, V:V). The PCBM solution was prepared by dissolving 10 mg of PCBM in 1 mL of mixed CB/ODCB (9:1, V:V). 1.2 M Cs$_{0.05}$FA$_{0.85}$MA$_{0.1}$Pb(I$_{0.95}$Br$_{0.05}$)$_3$ perovskite precursor solution was prepared in mixture solvent of DMF and DMSO (4:1, V:V). The p-type semicrystalline polymer (PDCBT, P3HT, PCDTBT, PTQ10, PTB7-Th, and PBDBT) was dissolved in ODCB with concentration of 15 mg mL$^{-1}$ and stirred at 90 °C. The BCF, Li-BCF, and C-BCF solutions were prepared by dissolving 2 mg Lewis acid into 1 mL IPA. The N-BCF and I-BCF solutions were prepared by dissolving 1 mg Lewis acid into 1 mL IPA. The dopant-free PTAA solution was prepared by dissolving 15 mg PTAA into 1 mL CB. The PTAA:BCF solution was prepared by dissolving 1 mg BCF into 1 mL PTAA CB solution (15 mg mL$^{-1}$).

### Device fabrication

For lateral-conducting device fabrication, the ready-made lateral interdigitated electrodes (purchased from Sichuan Aientropy Technology Co., Ltd) are cleaned by ultrasonic bath with acetone and ethanol before use. For solar cells fabrication, ITO substrates were sequentially cleaned by ultrasonic bath with acetone and ethanol, each cleaning step lasted for 30 min. The substrates were dried by blowing dry nitrogen stream and treated with O$_2$-plasma for 3 min. The SnO$_2$ layer was coated on the ITO glass by spin-coated SnO$_2$ solution at 4000 rpm for 30 s, followed by heating at 150 °C for 30 min in the air and transferred to glove box immediately. The PCBM solution was spin-coating on SnO$_2$ layer at 2000 rpm for 30 s and annealed at 150 °C for 10 min. The perovskite precursor was spin-coated onto the ITO/SnO$_2$/ PCBM substrates with three-steps program at 200 rpm for 3 s, 2000 rpm for 10 s, and 4000 rpm for 25 s, respectively. During the third step, CB was poured on the spinning substrate as antisolvent. The spin-coated perovskite precursor films were sequentially heated at 100 °C for 5 min and 150 °C for 10 min. The polymer HTL solution was deposited on the perovskite layers by spin-coating with 2000 rpm for 40 s with annealing at 90 °C for 10 min. Organic Lewis acid or Ta-WO$_x$ solution was coated on polymer at 3000 rpm for 30 s annealed at 80 °C for 5 min. Finally, Au electrode was deposited through a shadow mask via thermal evaporation.

### Characterization

J-V characteristics of PSCs were measured with a Keithley source measurement unit and a Newport Sol3A solar simulator under 100 mW cm$^{-2}$ AM1.5 G illumination. The incident light intensity was calibrated with a standard Si solar cell. An aperture mask with an area of 0.113 cm$^2$ was used. The EQE was measured through the Solar Cell Spectral Response Measurement System (Enli Technology Co., Ltd.). For the tracking of steady-state output, the PDCBT/BCF/Au-based device was operated under AM1.5 G illumination in ambient air at MPP based on reverse J-V scan for every 100 s (corresponding to Fig. 3e). The stabilized efficiency was obtained by multiplying the voltage bias by the current value. For the photo-thermal-operational stability characterization in the long term, the MgF$_2$ capping layers were deposited on Au electrodes, and devices were aged at elevated temperature in N$_2$-filled chambers under metal halide lamps equipped with UV filters having a cutoff at 400 nm. The MPP tracking is based on

reverse *J-V* scan every 1–3 h. The temperature is controlled by a hot-plate beneath the chamber.

*WF* measurement of polymers was performed with a SPS040 instrument from KP Technology. The tip's *WF* was determined by measuring the contact potential difference (CPD) between the tip and a standard gold reference sample for calculating the absolute *WF* values of samples. The ready-made lateral interdigitated electrodes (Sichuan Aientropy Technology Co., Ltd) are used for lateral conductance measurements. TOF-SIMS was performed by TOF.-SIMS 5 (ION-TOF GmbH, Germany). The in-situ UV-vis absorption spectra measurements were performed with TECAN Infinite 200 Pro. For long-term thermal stability testing of UV-vis absorption, the encapsulated films on a large common substrate were placed on a hotplate and kept at 55 °C in air, the substrates were transferred back and forth between the hotplate and analytical setup. The morphology and microstructure of the Au electrodes were characterized by field emission scanning electron microscope (SEM, HITACHI S4800). A cross-sectional lamella of PSCs, intended for TEM investigation, was prepared within a dual-beam FIB-SEM Helios NanoLab 660 (Thermo Fischer Scientific) by following the standard lift-out procedure. Vacuum-sealed solar cell device was unsealed just prior to the FIB processing. After the lamella preparation was completed, the sample was immediately transferred to TEM, limiting overall exposure to the ambient conditions less than 5 min. STEM imaging and EDS analyses were conducted using a double Cs-corrected Titan Themis (Thermo Fischer Scientific) TEM equipped with a SuperX detector and operated at 300 kV. EDS maps were acquired with a sampling size (i.e., pixel size) of 4.3 nm/pixel and a dwell time of 3 ms, utilizing a probe current of 80–100 pA. The acquired EDS maps were evaluated using TFS Velox software.

### Reporting summary

Further information on research design is available in the Nature Portfolio Reporting Summary linked to this article.

## Data availability

The data that support the findings of this study are available in the following repository: https://doi.org/10.6084/m9.figshare.24921918. The data generated in this study are provided in the Source Data file. Source data are provided with this paper.

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

## Acknowledgements

J.L., C.J., and Z.W. are grateful to the National Natural Science Foundation of China (Grant Nos. 62104031, 22175029, and 62374029). J.L. acknowledges the Technical Field Funds of 173 Project (Grant No. 24JJ210663A), the Natural Science Foundation of Shenzhen Innovation Committee (Grant No. CYJ20210324135614040), the Foundation of China Petroleum & Chemical Corporation (Grant No. 30000000–23-ZC0607-0127 36850000-23-ZC0607-0045), the Fundamental Research Funds for the Central Universities of China (Grant Nos. ZYGX2022J032 and ZYGX2021J010) for financial support. Z.W. acknowledges the Open Foundation of State Key Laboratory of Electronic Thin Films and Integrated Devices (Grant No. KFJJ202109). C.L. acknowledges funding from the European Union's Horizon 2020 research and innovation program under grant agreement No. 952911 ("BOOSTER") and 101007084 ("CITYSOLAR"). J.Z., K.Z., Z.X., and C.L. are grateful to the financial support from China Scholarship Council (CSC).

## Author contributions

J.L. and Y.Z. conceived the idea and designed the experiments. J.L., Z.W., N.L., L.L., C.J., Y.Z. and C.J.B. supervised the project. J.L., B.L., H.Y., H.S., and J.Y. fabricated the devices. J.L. T.H. and Y.Z. characterized device stability. K.Z. conducted EQE characterization. X.Z., M.W., and E.S. performed the STEM measurement. J.L., Y.Z. and C.L. performed the conductance and Kelvin probe measurements. J.Z. and J.W. performed the long-term stability of UV-vis absorption spectrum of polymers with various contacts. J.E. and Z.X. performed the SEM measurement. J.L. wrote the manuscript, C.J., Y.Z. and C.J.B contributed to the editing of this manuscript. All authors contributed to the discussion of the work.

## Competing interests

The authors declare no competing interests.
