## [Peer Review File · Nature Communications]

Polymer-acid-metal quasi-ohmic contact for stable perovskite solar cells beyond a 20,000-hour extrapolated lifetimeReviewer #1 (Remarks to the Author):

Review of "Polymer-acid-metal ohmic contact for stable perovskite solar 2 cells beyond a 15-year operational lifetime"

I believe this manuscript presents interesting results that warrant publication in a more energy focused journal, rather than a general journal like Nature Communications.

1. I do not believe it is suitable to put the term "...15-year operational lifetime" in a title. This is a bit subjective because it is determined by extrapolation and a solar simulator. Maybe "...15-year simulated operational lifetime"
2. From point 1, is the solar simulator in fact used for the stability testing? And if so, could the authors provide the spectra of the light source and indicate what is the amount of UV component?
3. Line 75 – I do not think the word "engenders" applies in this context.
4. Line 89 – should be accordingly.
5. The authors do mention cost at a few points in the manuscript, but do not provide actual numbers. The ultimate polymer selected for the study, PDCBT, seems to have a cost of 1g/\$3250 according to Ossila and 500mg/\$1986 Thermo-Fisher, which are not low cost. If the authors have other values 1-Materials please indicate.
6. The term crystalline polymers to describe the various polymers used in this study, but shouldn't a more appropriate term be semi-crystalline? These polymers are not fully crystalline.
7. Why do the authors select these polymers over something like PTAA? Did the authors study devices with a structure ...perovskite/PTAA/BCF derivative/gold? Thiophene containing polymers have a lower bandgaps than aromatic amine polymers like PTAA, thus can contribute to parasitic absorption of visible photons.
8. With regards to BCF, please indicate the actual name of these materials - tris(pentafluorophenyl)borane is BCF, lithium tetrakis(pentafluorophenyl)borate is Li-BCF, etc. Along these lines, BCF is clearly a Lewis acid, but I don't think the tetrakis(pentafluorophenyl)borate based molecules are considered Lewis acids since the empty boron orbital is no longer present. If the authors have another perspective on this, please indicate.
9. Line 130 – should be "studied"

Reviewer #2 (Remarks to the Author):

This paper reports the use of a Lewis-acid layer (BCF) between the p-type dopant free polymer (PDCBT) and metal (Au) in n-i-p PSCs. This interlayer creates a good ohmic contact for extracting photogenerated holes. The ohmic contact is maintained over a wide range of temperatures. The resulting device showed good performance (about 21%) for dopant-free n-i-p PSCs. The stability of these devices is good from 35 to 85 degrees. An activation energy of about 0.55 eV was derived. An estimated T80 of about 23012 to 24390 hours at 30 degrees suggests a service life of more than 15 years. Overall, this paper provides interesting insights for dopant free HTL work in n-i-p PSCs. However, this paper is not suitable for publication in its current stage due to a few important points that need to be addressed.

1. The activation energy and stability results are based on three individual cells, one cell at one temperature. This is not acceptable. It is known in the PSC field that there is often a large variation for PSC efficiencies. The variation of stability can be much larger. Thus, to draw any conclusion, a statistically significant comparison based on at least 5-10 devices at each temperature should be conducted. The average should be used to determine the stability and the activation energy. The error bars for the analysis should also be given.
2. 23012 to 24390 hours operation at 30 degrees cannot be equated to 15 years of service lifetime.

For practical outdoor operation, the conditions (temperature, illumination, etc.) can change dramatically over days, weeks, and months. Thus, to draw any connection to outdoor operation, the authors should conduct outdoor ageing tests.

3. It is interesting that the degradation activation energy is about 0.55 eV for the n-i-p PSCs used in this study. This value is very similar to a recent Nature article on p-i-n PSC stability, which showed an activation energy of about 0.6 eV. This similarity is interesting given the two studies used different contact layers and perovskite composition. Can the authors comment on this, with respect to the potential origin of the activation energy for both n-i-p and p-i-n PSCs.

4. Why can MgF₂ deposited on top of Au protect the device?

5. Is achieving good ohmic contact between the HTL and Au the most important factor for the improved stability? I think this is an overstatement about the importance of an ohmic contact. As for the state-of-the-art n-i-p PSCs, where spiro-OMeTAD is used, the stability is poor even at room temperature. However, I believe the contact is ohmic between spiro-OMeTAD and Au. This will contradict the conclusion from this study.

Reviewer #3 (Remarks to the Author):

The authors have substantially improved the stability of perovskite solar cells through the use of a Lewis acid to dope the polymer hole transport material. The authors have measured the rate of degradation at three different temperatures and calculated the acceleration factor. They estimate that the cells would operate for over 15 years at 30 degrees C. They call that the operational lifetime. I do not think that name should be used. The cells would frequently reach 40-55 degrees C in operation. Cells are often 30 C hotter than the ambient temperature. I am still impressed that an acceleration factor was determined. I have rarely seen anyone do that for perovskite solar cells. I recommend published this excellent manuscript after some minor edits are made. It would be better to refer to the polymers as semicrystalline rather than crystalline.

I recommend telling the reader that the Lewis acids are "coupled with alcohol-soluble Lewis acids (BCF/Li-BCF/C-BCF/N-BCF/I-BCF)" earlier in the manuscript. I don't think there is a sentence early in the manuscript pointing people to look at Figure 1.

Line 126: BCF should be spelled out.

Line 113: "disovle" is misspelled.

Line 181: "phonmena" is misspelled.

Line 208: "yielding a negligible hysteresis PCE of around 21%" is slightly confusing. I prefer "yielding a PCE of 21 % with negligible hysteresis."

**Point-by-point list of author actions in response to Reviewer comments**

Manuscript #: NCOMMS-23-47889

Black color: Reviewer's comment

Blue color: Author's response

Red color: Revision in the revised manuscript

**Reviewer #1**

Review of "Polymer-acid-metal ohmic contact for stable perovskite solar cells
beyond a 15-year operational lifetime".

I believe this manuscript presents interesting results that warrant publication in
a more energy focused journal, rather than a general journal like *Nature*
*Communications*.

**Response:** We sincerely thank the reviewer for this comment. In our work, we
underscore the pivotal role of a stable quasi-ohmic contact, a crucial element
not only within the realm of energy conversion but also across various
optoelectronic disciplines. By introducing a novel approach that combines a
Lewis acid layer with a semi-crystalline conducting polymer, we showcase a
new paradigm for achieving stable quasi-ohmic contacts.

We firmly believe that the implications of our research extend beyond the
confines of a specific field, and will be of substantial interest to the broad
readership of *Nature Communications*. Once again, we thank you for
recognizing the importance of our work and for considering its relevance to a
broader audience.

1. I do not believe it is suitable to put the term "...15-year operational lifetime"
in a title. This is a bit subjective because it is determined by extrapolation and
a solar simulator. Maybe "...15-year simulated operational lifetime".

**Response:** We thank the reviewer for this professional suggestion. We have
revised the title to “Polymer-acid-metal quasi-ohmic contact for stable
perovskite solar cells beyond a 20,000-hour extrapolated lifetime”.

**Page 1, Line 1:**

Polymer-acid-metal quasi-ohmic contact for stable perovskite solar cells
beyond a 20,000-hour extrapolated lifetime

2. From point 1, is the solar simulator in fact used for the stability testing? And
if so, could the authors provide the spectra of the light source and indicate what
is the amount of UV component?

**Response:** To clarify, the long-term stability assessment of our devices was
conducted under metal halide lamps equipped with UV filters having a cutoff at
400 nm. This information is supplemented in the revised manuscript.

**Page 23, Line 402:**

under metal halide lamps equipped with UV filters having a cutoff at 400 nm.

3. Line 75 – I do not think the word “engenders” applies in this context.

**Response:** We thank the reviewer for pointing out this issue. We have made
the necessary correction by replacing the word “engenders” with a more fitting
term, “gives rise to”.

**Page 4, Line 81:**

the significant similarity in solubility of the two polymers gives rise to a
processing challenge in which the upper polymer tends to dissolve the
underlying polymer.

4. Line 89 – should be accordingly.

**Response:** We feel sorry for this erratum. It is corrected in the revised
manuscript.

**Page 5, Line 97:**

Accordingly, the champion PSC exhibits excellent long-term photo-thermal-
operational stability under continuous illumination at temperatures of 35 °C (T_{99}
= 2,212 hours), 55 °C (T_{98} = 1,650 hours) and 85 °C (T_{82} = 937 hours) in N₂
atmosphere.

5. The authors do mention cost at a few points in the manuscript, but do not
provide actual numbers. The ultimate polymer selected for the study, PDCBT,
seems to have a cost of 1g/\$3250 according to Ossila and 500mg/\$1986
Thermo-Fisher, which are not low cost. If the authors have other values 1-
Materials please indicate.

**Response:** Thank you for highlighting the importance of addressing cost
considerations in our study. To address your inquiry, we have gathered the
prices for the specific materials referenced in the manuscript. The reference to
“lower cost” in our paper pertains to the comparison between PDCBT/BCF and
PDCBT/PTAA. We are optimistic about achieving even more cost-effective
outcomes by utilizing P3HT/BCF in future research, although the device
performance is still low. We appreciate your attention to this matter and assure
you that future iterations of our work will delve deeper into exploring more cost-
efficient and high-performance alternatives.

Product	Vendor	Price (\$/g)
PDCBT	1-Material	3000
PDCBT	Solarmer	2500
P3HT	Ossila	350
PTAA	Ossila	1850
BCF	TCI	110

**Page 14, Line 243:**

Thus, we successfully demonstrate an ideal quasi-ohmic contact with low cost
(Supplementary Table 1) and high reproducibility by using a polymer-acid-metal

architecture, providing an attractive alternative to commonly used metal oxides
and doped conducting polymers.

6. The term crystalline polymers to describe the various polymers used in this
study, but shouldn't a more appropriate term be semi-crystalline? These
polymers are not fully crystalline.

**Response:** We fully agree with the reviewer. We have corrected the "crystalline
polymer" to "semicrystalline polymer" throughout the whole manuscript.

7. Why do the authors select these polymers over something like PTAA? Did
the authors study devices with a structure ...perovskite/PTAA/BCF
derivative/gold? Thiophene containing polymers have lower bandgaps than
aromatic amine polymers like PTAA, thus can contribute to parasitic absorption
of visible photons.

**Response:** In our investigation, we observed that semicrystalline PDCBT films
exhibit efficient hole transportation, unlike amorphous PTAA films that
necessitate bulk doping for adequate hole transport. As evidenced in Fig. R1,
a device featuring the structure ITO/SnO₂/PCBM/perovskite/PTAA/BCF/Au
presents a low fill factor, highlighting the limitations associated with PTAA.

Regarding concerns about potential parasitic absorption due to PDCBT's light
absorption around 600 nm, it's noteworthy that this property becomes less
consequential when the perovskite layer exceeds 500 nm in thickness. This
mitigation is attributed to the high absorption coefficient of the perovskite film at
approximately 600 nm wavelengths.

**Fig. R1** The *J-V* curves of PSCs based on ITO/SnO₂/PCBM/perovskite/HTL/Au
 structure using the BCF surface processing. Red curve: HTL = PDCBT/BCF,
 blue curve: HTL = PTAA/BCF.

**Page 14, Line 246:**

Meanwhile, the performance of device with an architecture of
 ITO/SnO₂/PCBM/perovskite/PTAA/BCF/Au was also studied. In stark contrast
 to semicrystalline PDCBT film with efficient hole transporting, the amorphous
 PTAA film requires bulk doping to realize hole transporting. As a result, the
 device based on PTAA/BCF/Au architecture shows a low FF, as shown in
 Supplementary Fig. 15.

8. With regards to BCF, please indicate the actual name of these materials -
 tris(pentafluorophenyl)borane is BCF, lithium tetrakis(pentafluorophenyl)borate
 is Li-BCF, etc. Along these lines, BCF is clearly a Lewis acid, but I don't think
 the tetrakis(pentafluorophenyl)borate based molecules are considered Lewis
 acids since the empty boron orbital is no longer present. If the authors have
 another perspective on this, please indicate.

**Response:** We thank the reviewer for pointing out this issue. we have
 supplemented the revised manuscript with the full names of BCF and its

derivatives for clarity. We redefine that Li-BCF, C-BCF, N-BCF and I-BCF are
BCF's derivatives.

As you know, organic semiconductor doping is a complex issue because the
doping effect is dependent on the structural and electrical properties of both the
dopants and the organic semiconductors and on the interaction between them.

Peter K.H. Ho et al. and Thuc-Quyen Nguyen *et al.* confirmed the *p*-doping
effect by organic salts by using FET device and EPR/IR spectra, and they
further studied the mechanism of *p*-doping (*Nat. Commun.* 2016, 7, 11948; *ACS*
*Nano* 2018, 12, 3938). The downshift of the Fermi level is attributed to the
Hubbard/Coulomb interaction that attracts electrons from the conducting
polymers, as shown in Fig. R2. We hope this additional information sheds light
on the complexity of organic semiconductor doping and its intricacies related to
BCF derivatives and their impact on the Fermi level.

**Fig. R2** The mechanism of *p*-doping by organic salts of TrTPFB. Refer to *ACS*
*Nano* 2018, 12, 3938.

**Page 7, Line 126:**

tris(pentafluorophenyl)borane (BCF) and its derivatives (Fig. 1d), *i.e.*, lithium
tetrakis(pentafluorophenyl)borate-ethyl ether complex (Li-BCF),
triphenylmethylium tetrakis(pentafluorophenyl)borate (C-BCF), *N,N*-
dimethylanilinium tetrakis(pentafluorophenyl)borate (N-BCF) and 4-isopropyl-
4'-methyl-diphenyliodonium tetrakis(pentafluorophenyl)borate (I-BCF).

9. Line 130 – should be “studied”.

**Response:** We feel sorry for this erratum. It is corrected in the revised
manuscript.

**Page 9, Line 148:**

Interface contact was further studied based on the vertical architecture of
ITO/PDCBT/interlayer/Au.

**Reviewer #2**

This paper reports the use of a Lewis-acid layer (BCF) between the *p*-type
dopant free polymer (PDCBT) and metal (Au) in n-i-p PSCs. This interlayer
creates a good ohmic contact for extracting photogenerated holes. The ohmic
contact is maintained over a wide range of temperatures. The resulting device
showed good performance (about 21%) for dopant-free n-i-p PSCs. The
stability of these devices is good from 35 to 85 degrees. An activation energy
of about 0.55 eV was derived. An estimated T_{80} of about 23012 to 24390 hours
at 30 degrees suggests a service life of more than 15 years. Overall, this paper
provides interesting insights for dopant free HTL work in n-i-p PSCs. However,
this paper is not suitable for publication in its current stage due to a few
important points that need to be addressed.

**Response:** We thank the reviewer for the appraisal of our work and appreciate
his/her valuable time to give these helpful comments and suggestions.

1. The activation energy and stability results are based on three individual cells,
one cell at one temperature. This is not acceptable. It is known in the PSC field
that there is often a large variation for PSC efficiencies. The variation of stability
can be much larger. Thus, to draw any conclusion, a statistically significant
comparison based on at least 5-10 devices at each temperature should be
conducted. The average should be used to determine the stability and the
activation energy. The error bars for the analysis should also be given.

**Response:** We sincerely appreciate your feedback and the valuable insights
regarding the statistical significance of our stability and activation energy
analyses. We have revised the manuscript by including the trend of photovoltaic
parameters obtained from stability tests conducted on five representative
devices. These tests encompassed both reverse and forward *J-V* scans,
providing a more comprehensive understanding of device performance. The
error bar of photovoltaic parameters and activation energy are derived from the
five individual devices at different temperatures (Figs. R3-R16). These

additions enable a more robust analysis, resulting in a corrected activation
 energy of approximately 0.61 ± 0.11 eV for reverse J - V scans and 0.59 ± 0.10
 187 eV for forward J - V scans. All the new data are supplemented in the revised
 manuscript and supplementary information.

 **Fig. R3** Long-term stability of PSCs based on PDCBT/BCF/Au structure. **a,b**
 Data points originated from (a) reverse J - V scan and (b) forward J - V scan,
 respectively. **c,d** Normalized PCE originated from (c) reverse J - V scan and (d)
 forward J - V scan of PSCs plotted against the equivalent aging time at 30 °C by
 leveraging the Arrhenius relation. The error bars represent the standard
 deviations from 5 individual devices for each temperature.

 **Fig. R4** Logarithm of degradation rate (k) versus $1/k_B T$. **a,b** The data of k
 originated from analyzing J - V curves with (a) reverse and (b) forward scans of
 PSCs based on ITO/SnO₂/PCBM/perovskite/PDCBT/BCF/Au architecture. We
 used 5 individual devices for each stability test to obtain the average activation
 energy.

**Fig. R5** Normalized PV parameters of PSCs with structure of
 ITO/SnO₂/PCBM/perovskite/PDCBT/BCF/Au. **a-e** Normalized PV parameter of
 (a) device 1, (b) device 2, (c) device 3, (d) device 4 and (e) device 5 obtained
 from reverse *J-V* scans over time. The PSCs were aged at 35 °C in N₂-filled
 chamber under continuous light illumination (light intensity: 84 mW cm⁻²).

**Fig. R6** The statistic degradation behavior of PSCs with reverse J - V scan. **a-d**
 J_{sc} (a), V_{oc} (b), FF (c) and PCE (d) evolutions of PSCs based on
 ITO/SnO₂/PCBM/perovskite/PDCBT/BCF/Au architecture. The PSCs were
 aged at 35 °C in N₂-filled chamber under continuous light illumination (light
 intensity: 84 mW cm⁻²). The error bars were calculated from 5 individual devices.

**Fig. R7** Normalized PV parameters of PSCs with structure of
 ITO/SnO₂/PCBM/perovskite/PDCBT/BCF/Au. **a-e** Normalized PV parameter of
 (a) device 1, (b) device 2, (c) device 3, (d) device 4 and (e) device 5 obtained
 from forward *J-V* scans over time. The PSCs were aged at 35 °C in N₂-filled
 chamber under continuous light illumination (light intensity: 84 mW cm⁻²).

**Fig. R8** The statistic degradation behavior of PSCs with forward J - V scan. **a-d**
 J_{sc} (a), V_{oc} (b), FF (c) and PCE (d) evolutions of PSCs based on
 ITO/SnO₂/PCBM/perovskite/PDCBT/BCF/Au architecture. The PSCs were
 aged at 35 °C in N₂-filled chamber under continuous light illumination (light
 intensity: 84 mW cm⁻²). The error bars were calculated from 5 individual devices.

**Fig. R9** Normalized PV parameters obtained from reverse J - V scans over time
 for devices of ITO/SnO₂/PCBM/perovskite/PDCBT/BCF/Au. **a-e** Normalized PV
 parameter of (a) device 1, (b) device 2, (c) device 3, (d) device 4 and (e) device
 5. The PSCs were aged at 55 °C in N₂-filled chamber under continuous light
 illumination (light intensity: 84 mW cm⁻²) without encapsulation.

**Fig. R10** The statistic degradation behavior of PSCs with reverse J - V scan. **a-**
 **d** J_{sc} (**a**), V_{oc} (**b**), FF (**c**) and PCE (**d**) evolutions of PSCs based on
 ITO/SnO₂/PCBM/perovskite/PDCBT/BCF/Au architecture. The PSCs were
 aged at 55 °C in N₂-filled chamber under continuous light illumination (light
 intensity: 84 mW cm⁻²). The error bars were calculated from 5 individual devices.

**Fig. R11** Normalized PV parameters obtained from forward J - V scans over time
 for devices of ITO/SnO₂/PCBM/perovskite/PDCBT/BCF/Au. **a-e** Normalized PV
 parameter of (a) device 1, (b) device 2, (c) device 3, (d) device 4 and (e) device
 5. The PSCs were aged at 55 °C in N₂-filled chamber under continuous light
 illumination (light intensity: 84 mW cm⁻²) without encapsulation.

**Fig. R12** The statistic degradation behavior of PSCs with forward J - V scan. **a-**
 **d** J_{sc} (**a**), V_{oc} (**b**), FF (**c**) and PCE (**d**) evolutions of PSCs based on
 ITO/SnO₂/PCBM/perovskite/PDCBT/BCF/Au architecture. The PSCs were
 aged at 55 °C in N₂-filled chamber under continuous light illumination (light
 intensity: 84 mW cm⁻²). The error bars were calculated from 5 individual devices.

**Fig. R13** Normalized PV parameters of PSCs based on
 ITO/SnO₂/PCBM/perovskite/PDCBT/BCF/Au structure. **a-e** Normalized PV
 parameter of (a) device 1, (b) device 2, (c) device 3, (d) device 4 and (e) device
 5 obtained from reverse *J-V* scans over time. The unencapsulated PSCs were
 aged at 85 °C in N₂ under continuous light illumination (84 mW cm⁻²).

**Fig. R14** The statistic degradation behavior of PSCs with reverse J - V scan. **a-**
 **d** J_{sc} (**a**), V_{oc} (**b**), FF (**c**) and PCE (**d**) evolutions of PSCs based on
 ITO/SnO₂/PCBM/perovskite/PDCBT/BCF/Au architecture. The PSCs were
 aged at 85 °C in N₂-filled chamber under continuous light illumination (light
 intensity: 84 mW cm⁻²). The error bars were calculated from 5 individual devices.

**Fig. R15** Normalized PV parameters of PSCs based on
 ITO/SnO₂/PCBM/perovskite/PDCBT/BCF/Au structure. **a-e** Normalized PV
 parameter of (a) device 1, (b) device 2, (c) device 3, (d) device 4 and (e) device
 5 obtained from forward *J-V* scans over time. The unencapsulated PSCs were
 aged at 85 °C in N₂ under continuous light illumination (84 mW cm⁻²).

**Fig. R16** The statistic degradation behavior of PSCs with forward J - V scan. **a-**
 **d** J_{sc} (**a**), V_{oc} (**b**), FF (**c**) and PCE (**d**) evolutions of PSCs based on
 ITO/SnO₂/PCBM/perovskite/PDCBT/BCF/Au architecture. The PSCs were
 aged at 85 °C in N₂-filled chamber under continuous light illumination (light
 intensity: 84 mW cm⁻²). The error bars were calculated from 5 individual devices.

**Page 2, Line 42:**

our devices exhibit remarkable resilience, retaining $96\% \pm 3\%$, $96\% \pm 2\%$ and
$75\% \pm 7\%$ of their initial efficiencies after continuous operation in nitrogen
atmosphere at $35\text{ }^\circ\text{C}$ for 2,212 hours, $55\text{ }^\circ\text{C}$ for 1,650 hours and $85\text{ }^\circ\text{C}$ for 937
278 hours, respectively.

**Page 2, Line 46:**

we project an impressive T_{80} lifetime of 26,126 hours at $30\text{ }^\circ\text{C}$.

**Page 5, Line 97:**

the champion PSC exhibits excellent long-term photo-thermal-operational
stability under continuous illumination at temperatures of $35\text{ }^\circ\text{C}$ ($T_{99} = 2,212$
284 hours), $55\text{ }^\circ\text{C}$ ($T_{98} = 1,650$ hours) and $85\text{ }^\circ\text{C}$ ($T_{82} = 937$ hours) in N_2 atmosphere.

**Page 16, Line 274:**

The trend of normalized PCEs over time are shown in Figs. 4g and 4i after
calibrating for light intensity perturbations using a standard silicon
photodetector as reference. Five individual devices were measured under each
condition. The evolution of other photovoltaic parameters J_{sc} , V_{oc} and FF as a
function of aging time are summarized in the Supplementary Figs. 16-27.

**Page 16, Line 282:**

The devices retain $96\% \pm 3\%$, $96\% \pm 2\%$ and $75\% \pm 7\%$ of their initial PCEs
after continuous illumination at $35\text{ }^\circ\text{C}$ for 2,212 hours, $55\text{ }^\circ\text{C}$ for 1,650 hours
and $85\text{ }^\circ\text{C}$ for 937 hours with negligible J - V hysteresis.

**Page 17, Line 297:**

the fitted activation energy is $0.61 \pm 0.11\text{ eV}$ and $0.59 \pm 0.10\text{ eV}$ for the data
extracted from reverse and forward J - V scan, respectively.

**Page 18, Line 313:**

we estimate the T_{80} lifetime up to 26,126 and 22,795 hours at $30\text{ }^\circ\text{C}$ for PSCs
based on polymer-acid-metal structure (PDCBT/BCF/Au) in reverse and
forward J - V scan, respectively.

**Page 19, Line 331:**

The error bars represent the standard deviations from 5 individual devices for

each temperature.

**Page 19, Line 334:**

5 individual devices for each stability test were measured to obtain the average
activation energy.

**Page 20, Line 343:**

exhibit an exceptional degradation resistance at enhanced operational
temperatures, retaining $96\% \pm 3\%$, $96\% \pm 2\%$ and $75\% \pm 7\%$ of their initial
efficiencies during exposure to temperatures of $35\text{ }^\circ\text{C}$ for 2,212 hours, $55\text{ }^\circ\text{C}$ for
1,650 hours and $85\text{ }^\circ\text{C}$ for 937 hours, respectively.

2. 23012 to 24390 hours operation at 30 degrees cannot be equated to 15 years
of service lifetime. For practical outdoor operation, the conditions (temperature,
illumination, etc.) can change dramatically over days, weeks, and months. Thus,
to draw any connection to outdoor operation, the authors should conduct
outdoor ageing tests.

**Response:** Considering the reviewer's suggestion, we have deleted the
inappropriate description of outdoor operation throughout the manuscript.

3. It is interesting that the degradation activation energy is about 0.55 eV for the
n-i-p PSCs used in this study. This value is very similar to a recent Nature article
on p-i-n PSC stability, which showed an activation energy of about 0.6 eV. This
similarity is interesting given the two studies used different contact layers and
perovskite composition. Can the authors comment on this, with respect to the
potential origin of the activation energy for both n-i-p and p-i-n PSCs.

**Response:** It is a very interesting observation. We note that Zhu *et al.* adopted
a p-i-n structure of glass/ITO/MeO-2PACz/
$\text{Rb}_{0.05}\text{Cs}_{0.05}\text{MA}_{0.05}\text{FA}_{0.85}\text{Pb}(\text{I}_{0.95}\text{Br}_{0.05})_3/\text{C}_{60}/\text{SnO}_2/\text{Ag}$, reporting activation energy
of approximately 0.59 eV for the degradation rates in their latest *Nature*
publication (<https://www.nature.com/articles/s41586-023-06610-7>). Additionally,
we referenced Yueh-Lin Loo *et al.*'s work in *Science*, which explored a different

n-i-p structure of glass/FTO/TiO₂/Al₂O₃/CsPbI₃/Cs₂PbI₂Cl₂/CuSCN/Cr/Au,
reporting activation energy of approximately 0.24 eV and 0.43 eV for the
degradation rates with and without surface passivation, respectively. Given the
intrinsic stability of transparent conducting oxides (TCO) and other transporting
materials, our conclusions lean towards the influence of perovskite composition
on the observed activation energy similarities across different PSC structures.
Notably, our previous research on CsMAFA-perovskites also revealed an
activation energy of approximately 0.71 eV for decomposition (*Nat. Commun.*
*2021*, *12*, 2191). We expect a higher activation energy for the perovskites with
improved surface bonding, considering the initial degradation always starts with
the surface layer.

**Page 17, Line 298:**

The activation energy for our n-i-p structure is close to that reported in p-i-n
structure with a similar perovskite composition, implying that perovskite
decomposition determines the temperature-dependent degradation rate in
devices⁴⁴.

**Page 27, Line 531:**

44. Jiang, Q. et al. Towards linking lab and field lifetimes of perovskite solar
cells. *Nature* **623**, 313-318 (2023).

4. Why can MgF₂ deposited on top of Au protect the device?

**Response:** There is a high risk of breakage for Au electrode during the ageing
period at high temperatures, which is caused by trace vapors (e.g. I₂/MF/FA)
generated by perovskite film (Supplementary Fig. 30). This addition of MgF₂
serves to suppress the breakage susceptibility of the Au electrode, thereby
bolstering the overall stability of the PSCs under these conditions.

5. Is achieving good ohmic contact between the HTL and Au the most important
factor for the improved stability? I think this is an overstatement about the
importance of an ohmic contact. As for the state-of-the-art n-i-p PSCs, where

spiro-OMeTAD is used, the stability is poor even at room temperature. However,
I believe the contact is ohmic between spiro-OMeTAD and Au. This will
contradict the conclusion from this study.

**Response:** We acknowledge that while ohmic contact stands as a crucial factor,
it is not the exclusive determinant of enhanced stability. Actually, the ohmic
contact between spiro-OMeTAD and Au is found to be unstable during ageing
process at high temperatures, which will be reported in our forthcoming work.
It's essential to note that the overnight oxidation process used in spiro-
OMeTAD's fabrication primarily affects the interface rather than bulk doping,
contributing to the observed instability. This aligns with our ongoing
investigations, and we aim to present comprehensive insights into this matter
in our subsequent research.

**Reviewer #3**

The authors have substantially improved the stability of perovskite solar cells
through the use of a Lewis acid to dope the polymer hole transport material.
The authors have measured the rate of degradation at three different
temperatures and calculated the acceleration factor. They estimate that the
cells would operate for over 15 years at 30 degrees C. They call that the
operational lifetime. I do not think that name should be used. The cells would
frequently reach 40-55 degrees C in operation. Cells are often 30 C hotter than
the ambient temperature. I am still impressed that an acceleration factor was
determined. I have rarely seen anyone do that for perovskite solar cells. I
recommend published this excellent manuscript after some minor edits are
made.

**Response:** We thank the reviewer for the appraisal of our work and appreciate
his/her valuable time to give these helpful comments and suggestions. We have
revised the title to “Polymer-acid-metal quasi-ohmic contact for stable
perovskite solar cells beyond a 20,000-hour extrapolated lifetime”.

**Page 1, Line 1:**

Polymer-acid-metal quasi-ohmic contact for stable perovskite solar cells
beyond a 20,000-hour extrapolated lifetime

1. It would be better to refer to the polymers as semicrystalline rather than
crystalline.

**Response:** We appreciate the reviewer’s clarification. We have corrected the
crystalline polymers as “semicrystalline” polymers throughout the manuscript.

2. I recommend telling the reader that the Lewis acids are “coupled with alcohol-
soluble Lewis acids (BCF/Li-BCF/C-BCF/N-BCF/I-BCF)” earlier in the
manuscript. I don’t think there is a sentence early in the manuscript pointing
people to look at Figure 1.

**Response:** In light of the reviewer's suggestion, we have pointed out the
alcohol-soluble Lewis acid and its derivatives (BCF/Li-BCF/C-BCF/N-BCF/I-BCF)
early in the part of "Evolution of device architecture" in the revised manuscript.

**Page 7, Line 126:**

tris(pentafluorophenyl)borane (BCF) and its derivatives (Fig. 1d), *i.e.*, lithium
tetrakis(pentafluorophenyl)borate-ethyl ether complex (Li-BCF),
triphenylmethylmethyl tetrakis(pentafluorophenyl)borate (C-BCF), *N,N*-
dimethylanilinium tetrakis(pentafluorophenyl)borate (N-BCF) and 4-isopropyl-
4'-methyldiphenyliodonium tetrakis(pentafluorophenyl)borate (I-BCF) that are
dissolved in polar solvent such as isopropanol (IPA).

3. Line 126: BCF should be spelled out.

**Response:** We thank the reviewer for clarifying this. We have given the full
name of BCF in the revised manuscript.

**Page 7, Line 126:**

tris(pentafluorophenyl)borane (BCF).

4. Line 113: "disolve" is misspelled. Line 181: "phonmena" is misspelled. Line
208: "yielding a negligible hysteresis PCE of around 21%" is slightly confusing.
I prefer "yielding a PCE of 21 % with negligible hysteresis."

**Response:** We feel sorry for these errata. It is corrected in the revised
manuscript.

**Page 8, Line 130:**

are dissolved in polar solvent such as isopropanol (IPA).

**Page 12, Line 204:**

this phenomenon also observed in the pristine PDCBT layer.

**Page 13, Line 232:**

yielding a PCE of around 21% with negligible hysteresis.

Reviewer #1 (Remarks to the Author):

The authors have addressed my comments so the paper is fine to publish in my opinion.

Reviewer #2 (Remarks to the Author):

The authors have properly addressed my previous comments. The revised manuscript is suitable for publication at Nature Communications.

Reviewer #3 (Remarks to the Author):

I am fully satisfied with the responses to the comments and recommend publishing the manuscript.